# Evaluation of Three Different Vaccination Protocols against EHV1/EHV4 Infection in Mares: Double Blind, Randomized Clinical Trial

**DOI:** 10.3390/vaccines8020268

**Published:** 2020-06-01

**Authors:** Anna-Rita Attili, Renato Colognato, Silvia Preziuso, Martina Moriconi, Silvia Valentini, Stefano Petrini, Gian Mario De Mia, Vincenzo Cuteri

**Affiliations:** 1School of Biosciences and Veterinary Medicine, University of Camerino, 62024 Matelica, Italy; annarita.attili@unicam.it (A.-R.A.); silvia.preziuso@unicam.it (S.P.); martina.moriconi86@gmail.com (M.M.); 2Biologist, Senior Consultant, 20027 Ispra, Italy; renato.colognato@gmail.com; 3Veterinary Practitioner, 60010 Ostra, Italy; 4Veterinary Practitioner, 30028 San Michele al Tagliamento, Italy; valentinisilvia.vet@gmail.com; 5Istituto Zooprofilattico Sperimentale Umbria e Marche “Togo Rosati”, 06126 Perugia, Italy; s.petrini@izsum.it (S.P.); gm.demia@izsum.it (G.M.D.M.)

**Keywords:** horses, vaccines, herpesvirus, EHV1, EHV4, abortion, PCR, seroneutralization, ELISA

## Abstract

EHV1 and EHV4 are the most important herpesviruses in horses. Repeated cases of abortion in mares regularly vaccinated, prompted us to investigate the immune response after vaccination with the same inactivated vaccine, but with three different protocols. Eighteen mares were chosen and randomly divided in three study groups (G_1_-G_2_-G_3_) and a control group (Ctrl). For serologic and PCR investigations nasal swabs, sera and blood were collected. The protocol used in G_3_ (4 doses) increased the titer recorded by ELISA and seroneutralization (SN). Poor agreement and no correlation were observed in titer values between ELISA and SN and between SN and PCR. A very weak positive correlation between ELISA and PCR was obtained. Seven out of 18 nasal swabs were positive by PCR; none showed viremia and no abortion occurred, regardless of vaccination status and despite active circulation of EHV-1 in the farm at the time of the study. The study was conducted in field conditions, in a susceptible population with a known history of infection and abortion, and among the three protocols, the one proposed in the G_1_ was the least efficient while the one proposed for the G_3_, seems to have induced a higher antibody titer in both SN and ELISA.

## 1. Introduction

Herpesviruses, responsible for more or less severe diseases in equids, are viruses belonging to Alphaherpesvirinae and Gammaherpesvirinae subfamilies. The alpha herpes viruses are EHV1, EHV3, EHV4, EHV6 (also named AHV1), EHV8 (AHV3) and the last discovered EHV9; the gamma herpes viruses are EHV2, EHV5, EHV7 (AHV2), AHV4 and AHV5 [1,2]. Despite the long list, the recent discovery and the increasing knowledge of a new herpes virus related to EHV1, the most important viruses causing severe clinical disease and economic loss in equids are EHV1 and EHV4 [3], both included in the *Varicellovirus* genus. They are closely related, sharing genomic sequence homology [4,5,6] and common epitopes responsible for cross-reactivity [7,8]. Both are responsible for the worldwide spread of equine rhinopneumonitis [9,10]. Moreover, EHV1 can cause abortion as well as, less frequently, myeloencephalopathy [11,12]. Both viruses enter via the respiratory tract and replicate in the respiratory epithelium. After replication in the nasal mucosa, EHV1 infects the leukocytes of the tributary lymphoid tissue, which then results in a leukocyte-associated viremia. From there, viruses can spread reaching the end vessels of the pregnant uterus or those of the central nervous system (CNS) [3].

Based on the EHV1 pathogenesis, it has been suggested that three types of immune response are required for protection. These involve both the mucosal and systemic compartments, in which neutralizing antibodies acts against free virus particles, and lymphocytes capable to lyse virus infected cells. This cell associated viremia control process is a crucial point in abortion prevention and potentially also for the neurological disease onset as well [13].

The role of cytotoxic T lymphocyte (CTL) in protection against EHV1 has been the focus of intensive research. Especially quantification of the frequency of CTL precursors (CTLp) has demonstrated to be associated with immunity from experimental infection. Current evidence therefore supports the concept that an effective vaccine is needed to induce high titers of neutralizing antibody and elevated frequencies of CTLp both systemically and more important at the mucosal interface of the respiratory tract [14].

An expanded population of mucosa virus specific CD8+ memory CTL in the mucosa and in plasma will induce virus infected cells lysis, particularly lymphocytes, limiting viral spread to secondary replication sites such as pregnant uterus and/or spinal cord.

The seroneutralization (SN) and complement fixation (CF) tests are able to detect antibodies starting about 2 weeks after field or experimental infection with EHV1. Neutralizing antibodies are generally type-specific (i.e., EHV1 or EHV4), mainly directed against B (gB) and C (gC) glycoproteins and long-lasting, being able to persist for about 1 year after infection. In contrast, CF antibodies are short-lived, tending to disappear already 3 months after infection and cross react with both the EHV1 and EHV4 proteins, making them ineffective in differentiating the type of virus [13]. As regards the ELISA test, a kit has been developed that uses fusion proteins that express variable regions of gG and therefore allows to detect the type-specific antibodies of EHV1 and EHV4.

EHV1 and EHV4 infections are controlled primarily by sanitary measures that also include vaccination. Some commercial vaccines, with inactivated or modified live virus (MLV), have been shown to help protect from EHV-related respiratory disease, neurological disease and abortion under experimental conditions [15,16,17]. Many studies have described the serologic responses of on-field vaccinated horses [18,19]. Moreover, vaccine use is considered to be partially responsible for a reduction in the equine abortion incidence [20].

Despite vaccination, several cases of abortion caused by EHV1 occurred in a stable and prompted us to investigate the immune response of mares after vaccination with an inactivated commercial vaccine. Aim of our study was also to evaluate any difference in the immune response using different vaccination protocols. Sera were tested to detect antibodies against EHV1 and EHV4 by ELISA (SVANOVIR^®^ EHV1/EHV4-Ab) and SN. To evaluate the presence of virus shedding and/or viremia in vaccinated animals, blood and nasal swabs were used for PCR investigations.

## 2. Materials and Methods

### 2.1. Animals

Double-blinded, randomized, controlled field trial was carried out on eighteen trotter mares, between 6 and 10 years old, belonging to the same breeding located in north Italy, housed in the same barn and in a single box. The inclusion criteria were age less than 10 years, no clinical sign, body temperature less than 39 °C and no other general sign, like decreased appetite and/or lethargy. From 106 trotting mares 18 animals, representing the 17%, were randomly selected. All the animals were regularly vaccinated during the past years with the same vaccine (Duvaxyn EHV1/4, Fort Dodge, in the past and Equip EHV 1,4, Zoetis, nowadays) according to the manufacturer’s instructions and following the indicated schedule: pregnant mares, injected with a dose of vaccine, at the 5th, 7th and 9th month of pregnancy.

In the month preceding the beginning of the trial none of the mares were subjected to a previous laboratory investigation (PCR or serologic test) to find out their conditions against EHV1 and EHV4 infections and none of them were vaccinated or treated with anti-inflammatories or corticosteroids. The last vaccination was made in the previous breeding season.

After identifying the mares and completing the clinical record, 3 study groups (G_1_, G_2_, G_3_) of 5 mares and 1 control group (Ctrl) of 3 mares, were randomly created, using a random number generator (Random Generator for Microsoft Excel^®^, Microsoft Corporation, Redmond, Washington, USA).

In November 2013, upon request of a possible verification of the efficacy of the vaccine against EHV1/EHV4, the Ethics Committee for animal experimentation of the University of Camerino assessed and approved the experimental project (Minutes n. 11/2013).

All the animals used in this research were healthy or eventually naturally infected and any challenge was done. The vaccine used was commercially available and regularly registered for equine species and no suffering was caused to the animals during the evaluation.

### 2.2. Vaccine and Vaccination Protocols

The individual who performed the randomization was the same who administered the vaccinations and was not involved in any other component of the study. Laboratory staff were blinded to group assignments until all analyses were complete. The farm Veterinarian was in charge to carry out vaccinations on selected animals.

All the mares were injected with the same inactivated vaccine prepared with 438/77 EHV1 strain and 405/76 EHV4 strain and Carbopol 934P as adjuvant. G_1_ received the immunization at the third, fifth and seventh month of pregnancy; G_2_, at the fifth, seventh and ninth month of pregnancy as suggested by the Manufacturer; G_3_, 7 days before the expected date of birth (days before delivery, DBD) and at the first, fourth and sixth month of pregnancy. The Ctrl group have, instead, received sterile saline placebo at fifth, seventh and ninth month of pregnancy. The 2 mL vaccine dose and placebo were administered intramuscularly (IM) on the left side of the neck.

### 2.3. Samples Collection

For sample collection the blood was drawn by jugular venipuncture with a vacutainer collection system without coagulant (serum collection) or with heparin (for PBMC collection). An 18-gauge needle was used. Nasal swabs from each mare were also collected for molecular investigations. All samples were stored at −20 °C until further analysis. The farm veterinarian was in charge to collect the samples. No information was given to the veterinarian about the results of tests performed until the end of the work.

For the G_1_ group, all samples were collected at III (T_1_), IV (T_2_), V (T_3_), VII (T_4_) and IX (T_5_) month of pregnancy; for G_2_ at V (T_1_), VI (T_2_), VII (T_3_), IX (T_4_) and XI (T_5_) month of pregnancy; for G_3_ at I (T_1_), II (T_2_), IV (T_3_), VI (T_4_) and VIII (T_5_) month of pregnancy and for the Ctrl group at the V (T_1_), VI (T_2_), VII (T_3_), IX (T_4_) and XI (T_5_) month of pregnancy. Although T_2_ nasal swabs were collected, they were not analyzed because they did not reach the laboratory.

Since in some cases the vaccine was administered on the same day, blood samples and nasal swabs were collected before vaccination. Figure 1 shows the samples collection and vaccination protocols.

### 2.4. ELISA Test

The EHV1 and EHV4 IgG antibody quantification was performed by ELISA test (EHV-1/EHV-4 Discriminating Test, Svanovir, Svanova Biotech, Sweden). This test is based on the recombinant gG antigen of EHV1 and EHV4 [21] and is presented to be able to discriminate antibodies against EHV1 and EHV4 in both infected and vaccinated horses. The antibody values were detected by a 450 nm absorbance reading of each well. Positive values were taken from a cutoff of 0.2 according to the manufacturer’s instructions.

### 2.5. Seroneutralization Test

The standard seroneutralization test (SN) was performed as described in the OIE Manual for Terrestrial Animals, Equine Rhinopneumonitis, Chapter 3.5.9 (2008) [22].

Briefly, an aliquot of 25 µL of undiluted and inactivated serum and twofold dilutions of each were mixed with 25 µL of 100 TCID_50_ of EHV1 (Kentucky D strain) or EHV4 (405–75 strain) in a 96-well microtiter plates (SPL Life Sciences, Korea). Fifty microliters of rabbit kidney 13 cell culture (RK-13, ATCC, CCL-106) for EHV1 or Equine Derm cell culture (E-Derm, ATCC, CRL-6288) for EHV4 were added to each well. Positive and negative internal controls sera were used for every single test. Neutralization titers were expressed as log_2_ of the highest dilution inhibiting cytopathology.

### 2.6. PCR Assay

The PCR analysis was performed on both nasal swabs and blood samples. Blood buffy coat samples were obtained after centrifugation for 20 min at 2000× *g* at 4 °C. A Blood Genomic DNA Isolation Kit (Norgen Biotek Corporation, Thorold, ON, Canada) was used for blood DNA extraction while for nasal swabs samples the Genomic DNA Isolation Kit (Norgen Biotek Corporation, Thorold, ON, Canada) was used. As previously described [23] in order to detect EHV1 DNA, a nested-PCR protocol (nPCR), specific for the EHV1 gene ORF33 was performed. Briefly, first round nPCR mix contained 25 µL 2X Taq PCR Master Mix (Qiagen, Hilden, Germany), 1 µM of each primer (FC2-CTT GTG AGA TCT AAC CGC AC and RC-GGG TAT AGA GCT TTC ATG GG), 300 ng of DNA and PCR grade water up to 50-µL-final volume. Amplification conditions were 94 °C for 5 min followed by 35 cycles of 94 °C for 1 min, 60 °C for 1 min and 72 °C for 1 min, with a final extension at 72 °C for 7 min. The second round nPCR contained 25 µL 2X Taq PCR Master Mix (Qiagen, Hilden, Germany), 1 µM of each primer (FC3-ATA CGA TCA CAT CCA ATC CC and R1-GCG TTA TAG CTA TCA CGT CC), 2 µL of the first-round amplification products and PCR grade water up to 50 µL final volume. Amplification was carried out with the same temperature and time conditions used for the first round nPCR. Expected PCR size products after the first and second round PCR were 1118 bp and 188 bp, respectively. PCR products of both first and second round PCRs were visualized on 1.5% agarose gel containing ethidium bromide. To confirm their specificity, sequencing of PCR products was carried out (BMR Genomics, Padua, Italy).

### 2.7. Data Analysis

Statistical evaluation was performed by STATA software (S.E. version 13, College Station, TX, USA). By Kolmogorov–Smirnov test data were evaluated for normal distribution. After a first round of statistical analysis, we have observed that the data were not normally distributed, so a non-parametric testing analysis was subsequently used. Qualitative and quantitative variables were then evaluated for homogeneity among groups by a chi-squared and a Levene’s test, respectively. In order to analyze the effects of the different vaccination protocols for EHV1 and EHV4, ELISA and SN results were compared using chi-squared, both an intragroup evaluation, at the same time of collection and in the intergroup, at different times (T_n_–T_n+1_, T_first_–T_last_). The mean absorbance ELISA values were analyzed by Student’s *t*-test, both between groups and between intergroup. For the SN titer instead: (i) if quantitative data were not-paired, Wilcoxon’s stratified and Wilcoxon’s rank sum tests were used to compare intergroup, at different times and between groups, at the same time, respectively; (ii) if paired, the Friedman test by intergroup at different times was used. The Cohen’s Kappa test was used to evaluate the prevalence agreement between ELISA and SN, both for EHV1 and for EHV4. Finally, to correlate ELISA and SN titers with PCR results, Kendall (Tau) correlation was used. *p-*values <0.05 were considered statistically significant.

## 3. Results

Before vaccination, all groups showed homogeneity in the ELISA antibodies results for EHV1 (t = −0.415, *p* = 0.689), but not for EHV4 (t = 4.885, *p* = 0.0028), although they were all positive. About SN titers, homogeneity was observed only between groups (*R* > 17, *p* > 0.05).

### 3.1. EHV1

#### 3.1.1. Serology

Table 1 shows the ELISA data prevalence whereas the seroneutralization (SN) results are reported in Table 2. Significant differences in ELISA positive prevalence occurred at T_2_ between G_1_ and G_3_ (χ^2^ = 4.29, *p* = 0.0384), one month after the first vaccination (G_1_) and one month after the second vaccine dose (G_3_), between G_3_ and Ctrl (χ^2^ = 4.44, *p* = 0.0350) and at T_3_, 3 months after the second vaccination, between G_3_ and Ctrl (χ^2^ = 4.44; *p* = 0.0350). No significant difference, rising of the values, was instead recorded for SN prevalence percentages between groups (χ^2^ = 1.11, *p* = 0.292) and intergroup at different times (χ^2^ = 0.000, *p* = 1.000) (Table 1).

Mean ELISA values, recorded by groups at different times, are shown in Figure 2. No significant differences in ELISA and SN antibody titers (Table 1) were observed intergroup at different times, while significant difference by ELISA at T_5_ between G_1_ (9 months of pregnancy) and G_3_ (8 months of pregnancy) (t = −2.8396; *p* = 0.0296) was recorded (Figure 2).

Using a Wilcoxon’s rank sum test no statistical differences where observable between groups at the same time except for the SN titer recorded at T_3_: it was significantly greater in G_3_ (4 months of pregnancy) than in G_1_ (5 months of pregnancy) (*R* = 16.5; *p* < 0.05).

Considering the vaccinated groups as one group, by Wilcoxon’s stratified test emerged that the SN titers significantly decreased from T_1_ to T_2_ (*R* = 45, z = 3.31; *p* < 0.001). An increment statistically significant was instead observed from T_3_ to T_4_ (*R* = 52, z = 3.67; *p* < 0.001) and from T_4_ to T_5_ (*R* = 53, z = 3.13; *p* < 0.01).

Analyzing the quantitative SN intergroup data at different times, no differences were recorded by Friedman test (*F* < 3.32; *p* > 0.05).

Poor agreement (K = 0.0529, 5.29%) and no correlation were observed in titer values between ELISA and SN tests (mean Tau = 0.348; at T_1_: Tau = 0.091; at T_2_: Tau = 0.144; at T_3_: Tau = 0.386; at T_4_: Tau = 0.253; at T_5_: Tau = 0.714).

#### 3.1.2. PCR

A total of 7 nasal swabs (4 from mares of G_1_ and 3 from G_3_) collected at T_1_ were positive by PCR for EHV1, while all other nasal swabs and blood samples were negative. ORF33 sequences obtained from positive PCR samples showed 100% identity with each other and with nBLAST ORF33 EHV1 sequences (https://blast.ncbi.nlm.nih.gov/Blast.cgi).

No correlation was observed between SN and PCR values (mean Tau = 0.124; in G_1_: Tau = 0.5; in G_2_: Tau = 0; in G_3_: Tau = 0.1). A positive correlation, but not significant, was obtained between ELISA OD values and PCR (nanogram/µL) (r = 0.377, GL = 17; r^2^ = 0.1419285).

### 3.2. EHV4

#### Serology

Table 3 shows the SN results for EHV4. All horses showed a positive antibody response after vaccination. Figure 3 shows the significant decreases in titer values for G_1_ analyzing the T_1_ (3 months of pregnancy) and T_2_ (4 months of pregnancy) before the first vaccination and after one month (t = 3.787, *p* = 0.0053) and between the T_1_ (3 months of pregnancy) and T_5_ (9 months of pregnancy) sampling before the first vaccination and after 2 months the third dose of vaccine (t = 6.230, *p* = 0.0004). A similar significant trend was also observed for the G_2_ between the T_1_ (5 months of pregnancy) and T_2_ (6 months of pregnancy) (t = 3.6399, *p* = 0.0066) and between T_1_ (5 months of pregnancy) and T_4_ (9 months of pregnancy) (t = 3.002, *p* = 0.017) sampling. In contrast, after a decreased titer values observed in G_3_ between T_1_ (1 months of pregnancy) and T_2_ (2 months of pregnancy) (t = 7.5916; *p* = 0.0001), an increase was recorded from T_2_ (2 months of pregnancy) to T_3_ (4 months of pregnancy) (t = −2.7676; *p* = 0.0244) and from T_2_ to T_5_ (8 months of pregnancy) (t = −3.3744; *p* = 0.0118). In G_3_, at T_3_ (4 months of pregnancy) (t = −3.1986; *p* = 0.0126) and T_5_ (8 months of pregnancy) (t = −2.8994; *p* = 0.0274), the vaccination protocol allowed a significant greater titer compared to those ones observed in G_1_ (Figure 3).

By SN, among vaccinated groups, only in G_1_ and G_3_ a generalized rise of antibodies between T_1_ and T_2_ (*R* = 56; z = 3.19, *p* < 0.01), a decrease from T_3_ to T_4_ (*R* = 50; z = 3.92, *p* < 0.001) and a final significant rise from T_4_ to T_5_ (*R* = 26; z = 3.30, *p* < 0.001), were observed.

The intergroup analysis at different times did not show any statistically significant differences for SN titers (*F* < 3.32, *p* > 0.05). Between groups, at T_4_ the G_2_ had a greater SN titer than G_1_ (*R* = 15, *p* = 0.01) (9 and 7 months of pregnancy, respectively).

Nevertheless a perfect agreement was observed between ELISA and SN prevalence (K = 0.100, 100%), no correlation between ELISA OD values and SN titers was observed for EHV4 (mean Tau = 0.032; at T_1_: Tau = 0.085; at T_2_: Tau = 0.085; at T_3_: Tau = 0.248; at T_4_: Tau = 0.231; at T_5_: Tau = 0.214).

## 4. Discussion

The purpose of this study was to compare a standard vaccination protocol (proposed by the manufacturer) with two other alternative vaccination protocols using the inactivated Equip EHV 1.4 vaccine (Zoetis Italia S.r.l.). Our study hypothesis was prompted by repeated cases of abortion reported in mares regularly vaccinated with the vaccine under study and according to the protocol proposed by the manufacturer.

In particular, in the two years preceding the beginning of the study, there were at least 21 abortion cases diagnosed with EHV1 infection, despite a strict vaccination protocol in all the mares of the farm object of this study. During the study, following the administration of the vaccine, no cases of adverse reactions were reported, and no clinical symptoms related to herpetic infection were evident in all mares for both the vaccine treated or for the control groups. This fortuitous event was reached although seven (*n* = 7) out of eighteen (*n* = 18) animals were infected with EHV1 and virus was excreted via the respiratory system, as positive nasal swabs were recorded by PCR on samples collected at T_1_, before vaccination.

Other studies report that this vaccine is able to prevent abortion, reduce respiratory symptoms and duration of viral shedding amount [17].

In our study, there were also no abortions or respiratory signs, and virus excretion life that would have been evaluated only in horses tested positive in PCR at T_1_, cannot be calculated due to the fact that T_2_ nasal swabs did not reach the laboratory. Of course, this incident did not allow any assessment. However, viremia was not detected since the buffy coats were always PCR negative. This result would explain the absence of abortion, since it is known that it occurs only after an acute viremic phase. A further possibility could arise from the fact that the virus could have reached the target organs of latency after the excretion period, as reported by Patel and Heldens [3]. However, because during the period of the study, the veterinarian reported some cases of abortion in the rest of the animals, although not identified, it is not even possible to exclude that the vaccine worked well and protected the animals.

The evaluation of viral excretion by molecular techniques at the beginning of the study allowed to highlight some excretory animals (7 out of 18 animals). By simple randomization, 4 and 3 PCR positive animals were assigned in groups G_1_ and G_3_, respectively. The fact that no excretory animals existed in G_2_ and in the CTRL group allowed us to study animals placed in the recommended conditions for carrying out the following vaccination study. It also allowed to highlight that, on the basis of the SN results recorded at different times, the infection (present in the animals of groups G_1_ and G_3_) did not influence (no significant statistical differences were observed) the production trend of antibodies in groups G_1_ and G_3_ with respect to group G_2_, both for EHV-1 and for EHV-4.

Serologic results showed instead a difference for EHV1 prevalence in ELISA between G_1_ and G_3_ and between G_3_ and Ctrl at different time points. Results that were also reproducible for ELISA antibody titers at the end of the vaccination protocols between G_1_ and G_3_. These data show that the vaccination protocol used in G_3_, which includes four (*n* = 4) doses, could be better than others protocol (Figure 2). The seroneutralization (SN) antibody titers obtained are in agreement with this hypothesis since the SN titer recorded at T_3_ was significantly greater in G_3_ than in G_1_ (Table 2).

Considering all vaccinated animals as a whole group, SN titers decreased significantly from T_1_ to T_2_, while an increment was observed from T_3_ to T_4_ and from T_4_ to T_5_. These data suggest that the vaccine administration in animals with high antibody titers, due to immune memory following exposure to field virus or previous vaccination, could be responsible for an interaction between antibodies and vaccine, inducing a drastic lowering of protective antibodies. In contrast, when the antibody titer dropped, the vaccine can strengthen the immune system, resulting in a significant increase.

Poor agreement or no correlation in titer values between ELISA and SN tests for EHV1 was observed. This could be explained considering that the two serologic tests do not reveal the same antibodies. The same poor agreement or no correlation were observed between SN and PCR values. In this case, animals shedding viruses even if with high SN antibodies could be justified by the fact that neutralizing antibodies are not solely responsible for reducing the excretion of the virus, although in other study were shown that SN antibodies can reduce viral excretion from infected animals [24].

A very weak, but still to be considered correlation was instead obtained analyzing the ELISA test OD values and the PCR analysis (nanograms/µL) (r = 0.377, GL = 17; r^2^ = 0.1419285), respectively. With the increase in antibody titer, no viral excretion or only a slight increase was observed enough to hypothesize that the antibodies detected by the ELISA test are able to reduce the spread of the virus. However, further experimental infection studies should be performed to confirm this hypothesis.

The lack of DNA detection for EHV1 in almost all horses positive for EHV1 antibodies, could be due to low DNA quantity of EHV1. It is also possible that these serologically positive horses were latently infected and harbored EHV1 in classical (trigeminal ganglia and retropharyngeal lymph nodes) or “new” latently tissues types [25].

Gohering and colleagues [15] evaluated the ability of an MLV and an inactivated vaccine to control the virus excretion and viremia after experimental infection. Results showed a reduction of clinical signs and viral shedding after challenge, but disappointing these sequelae still developed despite an optimized vaccination regimen and challenge at the peak of the resulting immune responses.

As reported in other studies, serologic results for EHV4 confirmed that all horses were positive. This is probably due to gG ELISA sensitivity for the detection of the EHV1 antibodies which is lower than the specific EHV4 antibodies [26] or because EHV4 infection is quite frequent in the field.

Figure 3 shows the significant decreases in titer values in G_1_ before and after first vaccination (T_1_/T_2_), before the first vaccination (T_1_) and 2 months after the third dose of vaccine (T_5_) (ninth month). Similar significant trend was also observable for G_2_ before the first vaccination (T_1_) and after one month (T_2_), before the first vaccination (T_1_) and after 2 months and subsequently after the second dose (T_4_) (ninth month). In contrast, after a decreased titer value observed for G_3_, between one month after the first vaccination (T_1_) and one month after the second dose (T_2_), an increase was evident after one month subsequently the second vaccination (T_2_) at three months (T_3_) and from one month after the second vaccination (T_2_) to two months after the last vaccine inoculation (T_5_). For G_3_ at T_3_ and T_5_ sampling the vaccinations showed a significantly greater titer if compared to those observed for G_1_ (Figure 3).

By SN, among vaccinated groups, only for G_1_ and G_3_ it was observed a generalized increase of antibodies titer between T_1_ and T_2_, a decrease from T_3_ to T_4_ and a final significant rise from T_4_ to T_5_.

SN titers of each group, when compared at different times, did not show any statistically significant differences, instead, between the groups, only the G_2_ at T_4_ had a greater titer than those of G_1_.

An immune system stimulation was not observed following the vaccine administration—especially the typical booster effect after each vaccination. This failure of the humoral response may be explained by an inappropriate vaccine formulation, different antigen structure between the vaccine and the test virus or by the high antibody concentration before vaccination [27]. Another triggering cause could be the type of vaccine. In other studies, in fact, where an MLV and an inactivated vaccine were used, the SN assays showed that MLV vaccinated mares reached a higher SN titer when compared to those vaccinated with the inactivated vaccine, suggesting that the antibody response is also vaccine-type dependent [18,27]. In the same study [27], horses vaccinated against EHV in the past and/or with circulating antibodies anti-EHV appeared to be prevented from seroconversion. The same results are also observable in our study. In fact, in cases of high antibodies level, a drastic decrease following the vaccination was observed, potentially triggered by previous vaccination procedures or due to a farm-circulating field virus with consequent stimulation of the immune system.

The results obtained from our study show a clear difference in the immune response against the two types of viruses analyzed. In fact, at the first serum sampling, antibodies against EHV4 were higher than those against EHV1. This result seems to be potentially due to an intense circulation of both viruses in breeding rather than to previous vaccinations. This hypothesis is supported by the subsequent serologic test results, where, despite vaccinations, the expected typical booster effect, characteristic of each vaccination, has never been observed.

Our serologic results are similar to those presented by other authors, justifying the decrease in antibody titer after vaccine inoculation with a neutralizing action of the vaccine itself [19]. However, this hypothesis may be acceptable only for the first inoculation and it cannot be considered valid for the subsequent ones in which a progressive reduction of the antibody titer should be observed. This inability to increase the antibody titer by subsequent inoculations could also be explained by the presence of a high titer of anti-EHV4 antibodies, which could be responsible for the interference towards EHV1 due to the antigenic correlation [19].

Another hypothesis that may explain the booster effect failure could be founded in the work of Yasunaga and collaborators [28] where no evidence of an increase in antibodies after inoculation of the inactivated vaccine containing the strain EHV1-HH-1 BKS using a gG ELISA was reported. In the work, it was suggested that the virus in the vaccine is devoid of the sequence type-specific gG or that antibodies against gG were synthesized only after exposure to the field strain.

If it is true that the presence of antibodies is not a protection against infection and disease, which is instead attributed to cellular and local immunity [17], a fact to be considered is certainly the presence of viremia and/or viral excretion. In fact, no viremia condition was detected in any of the mares included in the study. The identification of viremic subjects, however, is not a common event. Viremia can be detected only during an acute infection, while it is highly unlikely to obtain a positive result from the buffy coat following reactivation of latent virus [29].

Another interesting consideration that can be done on the viral excretion is that, although observable only in T_1_, we found both in mares a high antibody titers (M_2_, M_3_ and M_5_ belonging to G_1_; M_1_, M_3_ and M_5_ belonging to G_3_), both in mares without antibodies (M_4_ of G_1_).

The failure to find the virus either from nasal swabs, except in some mares at T_1_ or from the buffy coat is also in agreement with what stated by Pusterla and collaborators [30] where in all samples collected before and after vaccination with a MLV or an inactivated vaccine observed negative EHV1 PCR results. These data of finding viruses in horses without clinical symptoms, vaccinated or not, could be a widespread event, as investigations on healthy subjects are rarely carried out and in many cases the infections of EHV1 and in particular of EHV4 are sub-clinical.

Positive PCR results indicate the presence of the virus. It is possible that this is not a latent virus, as the sample analyzed was neither the trigeminal ganglion, nor one of the lymphoid tissues associated to the respiratory tract. This may be a lytic virus, and our findings reflect a low-grade productive infection, despite vaccination.

## 5. Conclusions

In conclusion—despite the empiricism of the study and the low number of animals in any case similar to those used in other study [16]—among the different vaccination protocols, the one proposed in the G_1_ group that received the immunization at the third, fifth and seventh month of pregnancy, is the least efficient, while the one proposed for the G_3_ group, which consists of four doses, seems to have induced a higher antibody titer in both SN and ELISA. At the same time, it is very important to verify the serologic condition of the animals before vaccination because a good response is usually observed only in mares with a low ELISA titer at the initial vaccination with an inactivated vaccine [19]. Our study appears to have confirmed this event, since the presence of a high concentration of antibodies at the time of the first vaccination appears to have significantly influenced the immune response, inducing a rapid decline in antibodies, regardless of the protocol used.

No data on the protective ability of the vaccine against abortion could be obtained. The absence of abortion in the control group also confirms what was reported by other authors and that the level of antibodies is not the only absolute parameter for predicting clinical protection: cell-mediated and local immunity could also play an important role in providing protection [15,24].

Therefore, further investigations need to be performed in order to assess whether or not to use vaccination in mares during pregnancy and which protocol, among the various available, should be considered the most appropriate to reduce the occurrence of abortion.

## Figures and Tables

**Figure 1 vaccines-08-00268-f001:**
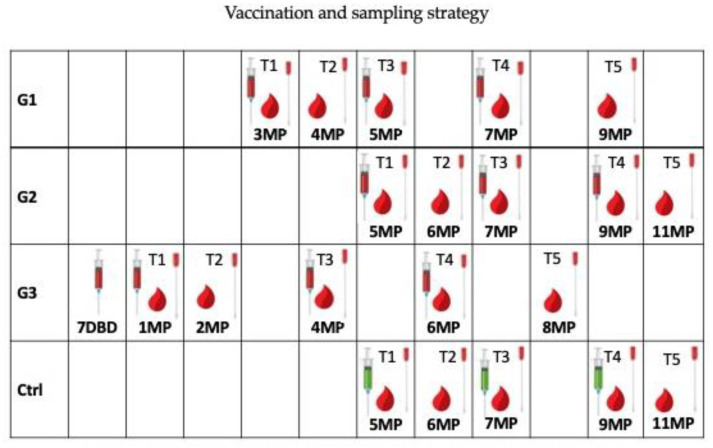
Experimental design—schematic representation of experimental layout. Red syringe = vaccine; green syringe = placebo; red drop = blood sample; swab = nasal swab; DBD = days before delivery; MP = month of pregnancy.

**Figure 2 vaccines-08-00268-f002:**
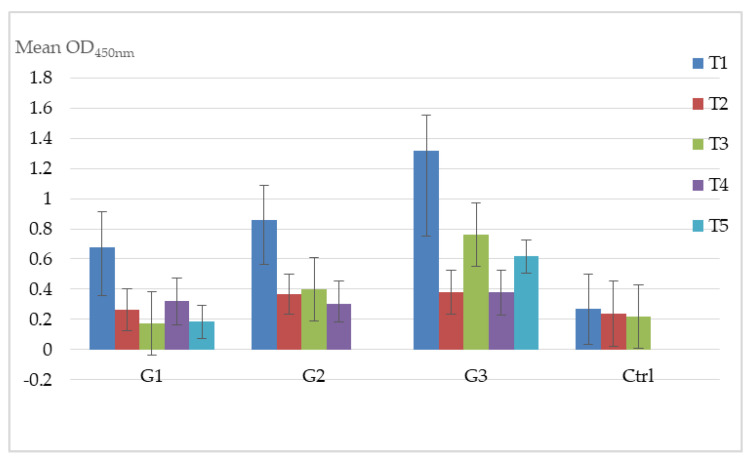
EHVI: Mean ELISA optical density (OD) values recorded by intergroup and between groups. Legend: T_5_ G_1_–G_3_ (t = −2.8396; *p* = 0.0296). Error bars: Standard Error of the Mean.

**Figure 3 vaccines-08-00268-f003:**
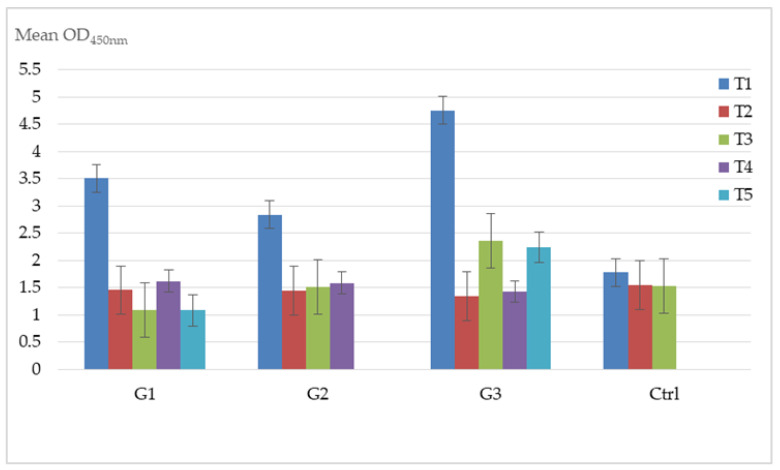
EHV4: Mean ELISA OD values recorded by intergroup and between groups. Legend: G_1_ T_1_–T_2_ (t = 3.787, *p* = 0.0053); G_1_ T_1_–T_5_ (t = 6.230, *p* = 0.0004); G_2_ T_1_–T_2_ (t = 3.6399, *p* = 0.0066); G_2_ T_1_–T_4_ (t = 3.002, *p* = 0.017); G_3_ T_1_–T_2_ (t = 7.5916; *p* = 0.0001); G_3_ T_2_–T_3_ (t = −2.7676; *p* = 0.0244); G_3_ T_2_–T_5_ (t = −3.3744; *p* = 0.0118); T_3_ G_3_–G_1_ (t = −3.1986; *p* = 0.0126); T_5_ G_3_–G_1_ (t = −2.8994; *p* = 0.0274). Error bars: Standard Error of the Mean.

**Table 1 vaccines-08-00268-t001:** ELISA and SN EHV1 prevalence (%): intergroup analysis at different times.

	G_1_			G_2_			G_3_			Ctrl		
	*MP*	*ELISA*	*SN*	*MP*	*ELISA*	*SN*	*MP*	*ELISA*	*SN*	*MP*	*ELISA*	*SN*
**T** _**1**_	3	80	80	5	80	100	1	100	100	5	33.3	100
**T** _**2**_	4	40 ^**1**^	80	6	60	100	2	100^**1**^	100	7	33.3 ^**2**^	100
**T** _**3**_	5	40	80	7	60	100	4	100 ^**3**^	100	9	33.3 ^**3**^	100
**T** _**4**_	7	40	100	9	60	100	6	75	100			
**T** _**5**_	9	50	75	11	60	100	8	100	100			

Legend: **^1^** = (χ^2^ = 4.29, *P* = 0.0384); **^2, 3^** = (χ^2^ = 4.44, *P* = 0.0350). SN: Seroneutralization test, MP: Month of Pregnancy.

**Table 2 vaccines-08-00268-t002:** EHV1: SN antibody titres.

EHV1		T_1_ (3MP)	T_2_ (4MP)	T_3_ (5MP)	T_4_ (7MP)	T_5_ (9MP)
G_1_	M 1	1:4	1:4	1:4	1:8	NA
	M 2	1:32	1:8	1:16	1:32	1:16
	M 3	1:32	1:8	1:16	1:32	1:32
	M 4	1:32	1:64	1:64	1:32	1:64
	M 5	1:64	1:64	1:32	1:64	1:64
G_2_		T_1_ (5MP)	T_2_ (6MP)	T_3_ (7MP)	T_4_ (9MP)	T_5_ (11MP)
	M 1	1:64	1:32	1:16	1:32	NA
	M 2	1:32	1:32	1:32	1:32	NA
	M 3	1:64	1:32	1:64	1:64	NA
	M 4	1:64	1:64	1:128	1:128	NA
	M 5	1:32	1:32	1:32	1:128	NA
G_3_ *		T_1_ (1MP)	T_2_ (2MP)	T_3_ (4MP)	T_4_ (6MP)	T_5_ (8MP)
	M 1	1:32	1:64	1:32	1:64	1:64
	M 2	1:64	1:32	1:128	1:128	1:64
	M 3	1:256	1:256	1:128	1:32	1:128
	M 4	1:32	1:32	1:128	1:8	≥1:512
	M 5	1:64	1:32	1:256	NA	NA
Ctrl		T_1_ (5MP)	T_2_ (6MP)	T_3_ (7MP)	T_4_ (9MP)	T_5_ (11MP)
	M 1	1:64	1:128	1:64	NA	NA
	M 2	1:32	1:32	1:8	NA	NA
	M 3	1:16	1:16	1:32	NA	NA

**Notes:** At T_1,_ T_3_ and T_4_ horses were vaccinated; *: G_3_ has received another vaccination at T_0._ M: mares, NA: not available, SN: Seroneutralization test, MP: Month of Pregnancy.

**Table 3 vaccines-08-00268-t003:** EHV4: SN antibody titres.

EHV4		T_1_ (3MP)	T_2_ (4MP)	T_3_ (5MP)	T_4_ (7MP)	T_5_ (9MP)
G_1_	M 1	1:16	1:16	1:8	1:16	NA
	M 2	1:32	1:32	1:16	1:16	1:16
	M 3	1:32	1:32	1:64	1:8	1:32
	M 4	1:32	1:32	1:64	1:16	1:64
	M 5	1:64	1:128	1:64	1:16	1:64
G_2_		T_1_ (5MP)	T_2_ (6MP)	T_3_ (7MP)	T_4_ (9MP)	T_5_ (11MP)
	M 1	1:64	1:128	1:64	1:64	NA
	M 2	1:64	1:128	1:64	1:32	NA
	M 3	1:128	1:128	1:128	1:128	NA
	M 4	1:64	1:64	1:128	1:64	NA
	M 5	1:128	1:64	1:64	1:64	NA
G_3_ *		T_1_ (1MP)	T_2_ (2MP)	T_3_ (4MP)	T_4_ (6MP)	T_5_ (8MP)
	M 1	1:64	1:64	1:32	1:64	1:32
	M 2	1:64	1:128	1:128	1:128	1:32
	M 3	1:64	1:64	1:128	1:128	1:128
	M 4	1:32	1:32	1:32	1:32	1:64
	M 5	1:64	1:128	1:64	NA	NA
Ctrl		T_1_ (5MP)	T_2_ (6MP)	T_3_ (7MP)	T_4_ (9MP)	T_5_ (11MP)
	M 1	1:64	1:64	1:128	NA	NA
	M 2	1:32	1:32	1:32	NA	NA
	M 3	1:32	1:16	1:32	NA	NA

**Notes:** At T_1,_ T_3_ and T_4_, horses were vaccinated; *****: G_3_ has received another vaccination at T_0_. M: mares, NA: not available, SN: Seroneutralization test, MP: Month of Pregnancy.

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
