# Peer review of "Evaluation of Three Different Vaccination Protocols against EHV1/EHV4 Infection in Mares: Double Blind, Randomized Clinical Trial"

_vaccines, 2020, doi:10.3390/vaccines8020268_

Round 1
Reviewer 1 Report
This is an interesting manuscript with and the data is well presented.
The study design is appropriate.
Reviewer 2 Report
The authors have improved the manuscript and have covered majority of my concerns. I have no further comments with regards to the manuscript.
Reviewer 3 Report
Authors have adequately addressed the concerns raised
Reviewer 4 Report
The authors have addressed all the questions raised by this reviewer.
This manuscript is a resubmission of an earlier submission. The following is a list of the peer review reports and author responses from that submission.
Round 1
Reviewer 1 Report
This is an insightful article on the development of veterinary vaccines.
The graphical abstract is very helpful to get the gist of the paper and draw readers' attention.
Please consider the following:
Addition of error bars in graphs. describe the results in a better fashion with more description.Author Response
Comments and Suggestions for Authors. Review n. 1
This is an insightful article on the development of veterinary vaccines.
The graphical abstract is very helpful to get the gist of the paper and draw readers' attention.
Please consider the following:
Addition of error bars in graphs. describe the results in a better fashion with more description.
All the figures were corrected, error bars were included, and a new figure was added.
Reviewer 2 Report
In summary
The authors have conducted series of serological tests to compare efficacy of inactivated vaccines in different vaccination regimens in apparent healthy trotter mares with no clear condition against EHV1 and EHV4 (lines 118-120).
The paper is written well to some extent and discusses the study from different angles.
There are some points to mention:
Major issues:
The study lacks an ethical statement. It is required that the authors consider to discuss ethical aspects of their animal study. This needs to be addressed completely and MUST be included in the materials/methods section. Please refer to below website for further information:https://www.mdpi.com/journal/vaccines/instructions#rethics
Please define what is delivery date/date before delivery (DBD)? Do the authors mean by DBD, days that the animals give birth? Or is there a different meaning? It is not clear that the horses are infected as a part of experimental procedure, or if they are left to be infected due to other reasons. Please clarify. In either case, the authors will need to include more information in the text. For example, if the horses are infected, the amount of virus will need to be included. Or if there is a different scenario, the scenario needs to be clearly discussed in the materials and methods. They are measuring nAb in their population without having the right set-up to compare protection against abortion, and in various places in the discussion, the authors are talking about “abortion”. I don’t think they can link abortion to their study. Hence, they can’t talk about it in the manuscript, because they didn’t set their experiment in such a way that they would study vaccination/infection effect to study abortion. In the discussion, the authors are mentioning that 7 out of 18 animals were infected with EHV1 during the study according to their PCR investigation. It is necessary to compare the PCR data with other techniques such as virus isolation to make sure that the band that they see is not non-specific. I am aware that the authors have done sequencing to confirm the nature of amplified DNA, but those data are not shown in the manuscript (or as a form of supplementary data). Did the investigators check animals prior infection status with EHV1/EHV4? In their result they talk about animals serological tests which may reflect on their previous state of infection. Authors need to discuss this clearly in their manuscript. Table 3- table legends. what is T0? It seems that the authors have human error in their study. For example, table 3, G3, M3, the antibody rise is not rational from T3 to T5. Similarly, for G3 M4. Why the graphs don’t have an error bar (figure 1 and 2). For displaying data, please consider displaying data based on group, not based on T. In the discussion, the authors should focus discussing the data. At its current form, the discussion, does not have a clear outline. The authors may want to arrange the discussion according to their results and focussing to explain the outcome of the study to support the conclusion. At its current form, there are many unrelated text in the discussion. Lack of consistency in writing is obvious throughout the paper (different fonts, etc: for example Figure 2). I have highlighted some of in this review. Considering the authors conclusion, the significance of study is not clear. Given the ethical issues around animal experiments, it is important that the authors to justify their study clearly in the manuscript.
Minor issues:
Table 1. Why authors use months of pregnancy (MP). In the text they refer to months of gestation (various places in text, for example lines 139-145).
Line 331: Laboratory: why capital “L” in laboratory
Line 427: it should read: ..similar to those presented ..
Line 412/449: it should read: … a high…
Table 4. What is ND?
Author Response
Comments and Suggestions for Authors. Review n. 2
In summary
The authors have conducted series of serological tests to compare efficacy of inactivated vaccines in different vaccination regimens in apparent healthy trotter mares with no clear condition against EHV1 and EHV4 (lines 118-120).
The paper is written well to some extent and discusses the study from different angles.
There are some points to mention:
Major issues:
The study lacks an ethical statement. It is required that the authors consider to discuss ethical aspects of their animal study. This needs to be addressed completely and MUST be included in the materials/methods section. Please refer to below website for further information:
https://www.mdpi.com/journal/vaccines/instructions#rethics
The ethical statement was included.
Please define what is delivery date/date before delivery (DBD)? Do the authors mean by DBD, days that the animals give birth? Or is there a different meaning?
Yes, DBD means Days Before Delivery, so 7 days before the presumptive date that the animals give birth. If you don’t like the word Delivery, we can change.
It is not clear that the horses are infected as a part of experimental procedure, or if they are left to be infected due to other reasons. Please clarify. In either case, the authors will need to include more information in the text. For example, if the horses are infected, the amount of virus will need to be included. Or if there is a different scenario, the scenario needs to be clearly discussed in the materials and methods.
The work was carried out under field conditions on a farm where Herpesvirus abortions had occurred. So some animals were naturally infected. No experimental infection has ever been performed.
They are measuring nAb in their population without having the right set-up to compare protection against abortion, and in various places in the discussion, the authors are talking about “abortion”. I don’t think they can link abortion to their study. Hence, they can’t talk about it in the manuscript, because they didn’t set their experiment in such a way that they would study vaccination/infection effect to study abortion.
The observation is appropriate. Our goal was to evaluate the antibody situation following different vaccination protocols, not to verify whether the vaccine was able to protect against abortion. The vaccine is a commercially available product. When we state about abortions in our paper, we always link it to the fact that PCR has always showing a negative blood results, demonstrating the absence of viremia. Abortions are known to occur in the presence of viremia. So, we only justify the absence of clinical signs in horses vaccinated for the absence of viremia and not due to the fact that they have been vaccinated.
In the discussion, the authors are mentioning that 7 out of 18 animals were infected with EHV1 during the study according to their PCR investigation. It is necessary to compare the PCR data with other techniques such as virus isolation to make sure that the band that they see is not non-specific. I am aware that the authors have done sequencing to confirm the nature of amplified DNA, but those data are not shown in the manuscript (or as a form of supplementary data). Did the investigators check animals prior infection status with EHV1/EHV4? In their result they talk about animals serological tests which may reflect on their previous state of infection. Authors need to discuss this clearly in their manuscript.
e performed PCR on all horses before starting vaccination protocols and 7 out of 18 showed positive results. Sequencing analysis was also performed on all positive samples and EHV1 was always confirmed. The data has been added to the text.
Table 3- table legends. what is T0?
T0 is reported as the time before pregnancy (7 days before childbirth). We only used this term to differentiate it from other time points used at different pregnancy times.
It seems that the authors have human error in their study. For example, table 3, G3, M3, the antibody rise is not rational from T3 to T5. Similarly, for G3 M4. Why the graphs don’t have an error bar (figure 1 and 2). For displaying data, please consider displaying data based on group, not based on T. In the discussion, the authors should focus discussing the data. At its current form, the discussion, does not have a clear outline. The authors may want to arrange the discussion according to their results and focussing to explain the outcome of the study to support the conclusion. At its current form, there are many unrelated text in the discussion. Lack of consistency in writing is obvious throughout the paper (different fonts, etc: for example Figure 2). I have highlighted some of in this review. Considering the authors conclusion, the significance of study is not clear. Given the ethical issues around animal experiments, it is important that the authors to justify their study clearly in the manuscript.
The Error bars were included within the figures data. As suggested Figures were correct. Ethical approval was included.
Minor issues:
Table 1. Why authors use months of pregnancy (MP). In the text they refer to months of gestation (various places in text, for example lines 139-145).
Line 331: Laboratory: why capital “L” in laboratory
Line 427: it should read: ..similar to those presented ..
Line 412/449: it should read: … a high…
Table 4. What is ND?
Corrected in the text
Reviewer 3 Report
AUTHORS
Article ID: vaccines-694124
Title: EHV1/EHV4 infection in mares: a preliminary evaluation of three different vaccination protocols.
This is a study evaluating the immune response after vaccination with an inactivated vaccine and to evaluate any differences with different protocols. This was performed due to repeated cases of abortion in mares regularly vaccinated. Authors tested 3 different vaccine protocols in 3 groups of mares plus one control group. Serological and PCR follow-up was performed. Among the different vaccination protocols, the one consisting in four doses would seem to be better in terms of antibody concentration. Interesting study however I have a few questions that need clarification.
Please define seroneutralization (SN) when first used on abstract.
You later use “virus neutralization (VN)
What antibody classes are evaluated with the ELISA and the SN?
Authors evaluated which animals were positive by PCR before starting the study? Or only during the study? I am not an EHV expert but could active EHV infection influence the antibody response to that EHV? How antibody rise can be discriminated as coming from infection or vaccine?
Could SN and ELISA distinct results be explained by IgM or IgA increases due to more frequent exposure to EHV and not to vaccine strain?
Was there any ethics committee permission requested for this study?
Author Response
Comments and Suggestions for Authors. Review n. 3
This is a study evaluating the immune response after vaccination with an inactivated vaccine and to evaluate any differences with different protocols. This was performed due to repeated cases of abortion in mares regularly vaccinated. Authors tested 3 different vaccine protocols in 3 groups of mares plus one control group. Serological and PCR follow-up was performed. Among the different vaccination protocols, the one consisting in four doses would seem to be better in terms of antibody concentration. Interesting study however I have a few questions that need clarification.
Please define seroneutralization (SN) when first used on abstract. Corrected in the text
You later use “virus neutralization (VN). Corrected in the text
What antibody classes are evaluated with the ELISA and the SN? IgG were always evaluated.
Authors evaluated which animals were positive by PCR before starting the study? Or only during the study?
PCR was performed on all animals before starting the vaccination protocol and during the whole study.
I am not an EHV expert but could active EHV infection influence the antibody response to that EHV?
No positive blood results were obtained by PCR, and any viremia was revealed. This appears not to have affected the antibody response.
How antibody rise can be discriminated as coming from infection or vaccine?
The used vaccine is not a DIVA, so it is not possible to discriminate if antibodies are from the infection itself or from the vaccination.
Could SN and ELISA distinct results be explained by IgM or IgA increases due to more frequent exposure to EHV and not to vaccine strain?
Both tests are able to detect only IgG, and neither IgM or IgA.
Was there any ethics committee permission requested for this study?
Yes, it was included.
Reviewer 4 Report
I recommend that Table 1 be modified to better show what was done with group 1, 2, and, which were described in 2.1 to 2.3. The group numbers should be put in the headings.
Lines 220-221. It is unclear what SN titer homogeneity they see between groups. What groups? Based on Table 3, I do not see any similarities between groups (G1-3).
Line 230. What is G4, which was compared with G3?
Line 241. “Mean OD values…” should read “Mean ELISA values…”
Figure 1. Error bars are missing. How many animals each bar represents? Why did they not consider the difference between G1 and G3 significant at T1 and T3 as well? Why only at T5? What is G4, maybe the control group? The labeling should be consistent throughout the paper. Why are some measurements missing at T4 and T5? The title of the Y-axis is missing.
3.1.2 PCR. There were 18 horses in the study but only 4 of G1 and 3 of G3 were tested by PCR. No animals were tested from G2 and the control group? I do not think that this PCR section has any value in the manuscript. It should be removed or add the PCR data of all horses.
Line 296-299. I do not see a general (not generalized) rise of antibodies between T1 and T2 in groups G1 and G3. Only 1 or 2 out of 5 horses show titer increase. Actually, the same is true for G2 as well.
It is unfortunate that no antibody titer data is available for some horses at T4 and many of them at T5 (Table 4), which could better show the antibody titer change from T1 to T5.
Figure 2. Error bars are missing. How many animals were tested per group? What is G4? The title of Y-axis is missing.
Author Response
Comments and Suggestions for Authors. Review n. 4
I recommend that Table 1 be modified to better show what was done with group 1, 2, and, which were described in 2.1 to 2.3. The group numbers should be put in the headings.
Modified with a new figure according to your suggestion.
Lines 220-221. It is unclear what SN titer homogeneity they see between groups. What groups? Based on Table 3, I do not see any similarities between groups (G1-3). ??
The reason for which we state that there is a similarity between groups is derived by the fact that we observed in G1 a SN titer lower than those observed for G2, G3, and Ctrl although not statistically significant. Under these circumstances we can only affirm that the groups are similar.
Line 230. What is G4, which was compared with G3? Corrected in the text. Is the renamed Ctrl group.
Line 241. “Mean OD values…” should read “Mean ELISA values…” Corrected in the text
Figure 1. Error bars are missing. How many animals each bar represents? Why did they not consider the difference between G1 and G3 significant at T1 and T3 as well? Why only at T5? What is G4, maybe the control group? The labeling should be consistent throughout the paper. Why are some measurements missing at T4 and T5? The title of the Y-axis is missing.
Modified with a new figure according to your suggestion.
3.1.2 PCR. There were 18 horses in the study but only 4 of G1 and 3 of G3 were tested by PCR. No animals were tested from G2 and the control group? I do not think that this PCR section has any value in the manuscript. It should be removed or add the PCR data of all horses.
We changed the sentence because it was not clear. As mentioned in the materials and methods, we tested all swab and blood samples by PCR but only seven samples collected at T1 resulted positive.
Line 296-299. I do not see a general (not generalized) rise of antibodies between T1 and T2 in groups G1 and G3. Only 1 or 2 out of 5 horses show titer increase. Actually, the same is true for G2 as well.
This data could be better appreciated with the new figures.
It is unfortunate that no antibody titer data is available for some horses at T4 and many of them at T5 (Table 4), which could better show the antibody titer change from T1 to T5.
The farm where the study was performed was more than 600 km away from the laboratory and it was necessary to use an express courier for the shipment. We were particularly unlucky, in one case the package was lost and in a second case the tubes containing the samples were delivered broken. In addition, some animals were sold, and follow-up was not possible.
Figure 2. Error bars are missing. How many animals were tested per group? What is G4? The title of Y-axis is missing.
Corrected with a new figure.
Round 2
Reviewer 2 Report
Regarding the word delivery, it is not that I don't like the term "delivery". It has to be mentioned clearly in the text what is your definition from "delivery. Please consider changing it in the manuscript to avoid confusion.
The authors statement is correct, abortions are linked with the presence of viremia. And it is linked in the manuscript.
Author Response
Many thanks for your further suggestions.
The word delivery has been better defined in the text.
All highlighted sentences have been revised.
Best regards
Vincenzo Cuteri